# Elaboration of massage technique for semen collection and examination of semen characteristics in chinchilla (*Chinchilla lanigera*)

**Bianka Babarczi**[1]*, **Árpád Drobnyák**[2], **Judit Barna**[2], **Éva Váradi Kissné**[2], **Zsuzsa Szabó**[2], **Mónika Heincinger**[2], **Károly Kustos**[2], **Zsuzsanna Szőke**[1], **Barbara Végi**[2]

**1** Hungarian University of Agriculture and Life Sciences, Institute of Genetics and Biotechnology, Department of Animal Biotechnology, Reproductive Bilogy and Toxicology Group, Gödöllő, Hungary, **2** National Centre for Biodiversity and Gene Conservation, Institute for Farm Animal Gene Conservation, Institute for Gene Conservation Science and Small Animal Research, Gödöllő, Hungary

* bianka421@gmail.com

**Data Availability Statement:** All relevant data are within the paper.

## Abstract

The practice of artificial insemination for the long-tailed chinchilla has not been fully elaborated to date, and existing data available regarding their reproduction properties is contradictory. Until now, the collection of semen for chinchillas has been most-commonly obtained using electro-ejaculation methods exclusively. The primary objective of this study was the development of a manual technique for semen collection which meets all animal welfare requirements. An additional aim was to determine the basic spermatological parameters, such as motility, concentration, type and ratio of morphological abnormalities and live/dead cell ratio, under typical northern-hemisphere conditions, in Hungary. Over a 3 month period, a special massage technique was developed for the study, and using this method, the sperm parameters of 46 males were subsequently analyzed weekly for a period of one year. Approximately 66% of chinchillas responded positively to this technique, with the success rate of semen-collection attempts showing no variation between seasons. Average sperm concentration for the whole year was 935.17 million/ml using this method. Total cell motility was the highest in winter (90.3%), and the lowest in spring (84.3%). The proportion of live, intact cells were above 80% on average for the year, while the ratios of live, morphologically abnormal and dead cells were 6% and 14%, respectively. We found that midpiece abnormalities occurred in the highest proportion (0.95%-3.38%), while the head abnormalities showed the lowest ratio (0.01%-0.15%). Standard deviation among the parameters was relatively high, with the spring season proving to be the weakest in terms of sperm quality. This study has demonstrated that, semen can be successfully collected without the use of electro-ejaculation or anesthesia. Furthermore, although spermatological parameters do exhibit some fluctuation for the different times of the year, semen collected is nonetheless suitable for the purpose of artificial insemination of chinchillas at any time.

**Funding:** This study was supported by the Operational Programme of Economic Development and Innovation / Hungary (Project code: GINOP-2.1.7-15-2016-02232). The funders had no role in study design, data collection and analysis, decision to publish, or preparation of the manuscript.

## Introduction

Chinchillas in the wild are currently on the verge of extinction; with only a few smaller-sized colonies living in the Andes [1]. As the chinchilla (*Chinchilla lanigera*) is an endangered species, any practical research on their reproduction can be considered to be of paramount importance.

In addition, studies on the reproductive traits of captive chinchillas may also be beneficial from a commercial and economic point of view as well. An appropriate *ex situ*, *in vivo* breeding program and the development of assisted reproduction techniques could appreciably support the management of chinchilla populations [2]. In the breeding farms, artificial insemination could be a breakthrough for chinchilla breeding, mainly due to the intensive selection and use of males that produce the best fur quality. But in order to achieve this, it is first essential to understand the gamete physiology for the species [2]. Before all else, it is necessary to establish a method of semen collection which can be performed in a routine manner, taking into account all applicable animal welfare considerations. In other animal species the most common sperm collection techniques are artificial vagina, digital manipulation and electroejaculation [3]. Until now, the sperm collection method exclusively has been electro-ejaculation in chinchilla, with or without anesthesia; however, aside from not meeting current animal welfare criteria, this practice is also not the most effectual nor reliable in terms of sperm quality. Dalziel and Phillips [4] were the first to apply electro-ejaculation techniques to guinea pigs and chinchillas in 1948. Later, Healey and Weir [5, 6] also published a detailed electro-ejaculation technique for chinchilla. However, it is now considered as fact that this procedure is both highly painful as well as stressful for the animals involved. Several studies have shown that electro-ejaculation induces physiological, neuroendocrine, and behavioral changes that suggest a stress response associated with pain [7–10]. As a result of the findings referenced above, the use of electro-ejaculation without anesthesia has already been banned in several European countries [11]. Moreover, even from a practical standpoint, this technique is neither the most effective nor the simplest to administer, as anesthesia of animals is also required for its implementation [12]. During electro-ejaculation, the semen frequently becomes mixed with urine, which has an extremely adverse effect on the semen [13]. Unfortunately, for many animal species, it is still the most common method of obtaining semen. Furthermore, previous research on chinchilla demonstrated that electro-ejaculation could only be performed as frequently as once every 2 weeks, or sometimes once a week [2, 14, 15].

Digital manipulation or massage technique is routinely used for sperm collection in pigs, dogs [3, 16], and most of avian species [17]. The method of digital manipulation is effective, easy to perform. [18], males can be induced to ejaculation by employing pressure and massage to the penis [3].

Hence, the research and understanding of the reproductive biological properties of chinchilla is of the highest importance, especially with regard to the development of practical methods and protocols for the effective collection, management and conservation of semen. From the standpoint of the males, the success of semen collection is greatly influenced by the use of a gentle, animal-friendly method. The first step is determining a method of the semen collection which is able to be performed in a routine manner, taking into account current animal welfare considerations [7–10, 19].

Furthermore, in order to study the reproductive biological characteristics of the male animals, the measurement and determination of basic spermatological parameters is essential. Data available on these [2, 10, 14, 15, 20, 21], has proven to be either incomplete, or wide-ranging and mostly contradictory.

Our study is the first to examine the possibility of using a massage method for sperm collection in chinchilla which is both safe, as well as makes the routine, long-term collection of semen possible throughout the whole year. An additional consideration of this study was to determine the basic spermatological parameters for a complete yearly cycle, under the northern-hemisphere conditions of Hungary.

# Methods

## Animals

In the study, 46 sexually mature, one-year-old male chinchillas were used. Animals were selected randomly, kept on a 12L:12D lighting program and temperature between 17–25°C, depending on the season. They were fed with pelleted chinchilla food, had access to water *ad libitum*, and received a cube of compressed alfalfa once weekly.

Animals were housed individually in stainless steel cages. A spoon of sand-bath powder was added to each cage on a regular basis so the animals could perform a "dust bath" to keep their fur dry and uncompressed.

## Semen collection

Following an anatomical study of the mating organ of the males, it took about 3 months of experimentation to develop the proper massage technique. The technique was based on digital manipulation. Collection of semen from the chinchilla using massage technique don't requires sedation and special restraint. The procedure is as follows: one person holds the male on his left forearm with his back to the sperm collector person while the other hands holds an Eppendorf tube containing 500μl of PBS medium and a single layer of gauze sheet in order to catch the gel-like copulatory plug of seminal plasma. The collector lifts the animal's tail with one hand and gently massages the penis with the other hand along the entire length of the penis until ejaculation (Fig 1). Removing the gauze, a gentle pressure was applied to it in order to achieve as much sperm cells as possible. Then the gauze was discarded together with the gel-like coagulate plug. Sperm collection was performed twice a week, and the sperm parameters of 46 males were analysed weekly for a period of one year.

## Sperm functional activity

**Sperm motility and concentration.** Spermatological qualification was performed using the CASA system (Microptic S.L. SCA®). For this study, collected semen was diluted to such an extent (0-20x dilution in 0.9% saline solution) that program was able to analyse individual movements. 10μl of the sample was applied to a heated slide and multiple fields of view were used to effectively analyse the motility of 500 cells.

Using this methodology, the ratio of rapid progressive, medium progressive, nonprogressive and immotile cells could be reliably determined. In addition, the total motile cell ratio and concentration of each semen sample was also determined by the CASA system.

**Sperm viability.** Sperm viability was evaluated by aniline blue-eosin staining (Fig 2). This method is based on eosin (Certistain, 115935 Eosin Y, Merck Ltd., Budapest, Hungary), by which damaged sperm membranes become stained, while live, intact cell membranes remain impermeable. Aniline blue (415049 Sigma-Aldrich Ltd., Budapest, Hungary) creates a blue background in the smear, enhancing the contrast of the white, unstained sperm having intact cell membranes, along with allowing morphological abnormalities to also become clearly visible. 20 μl of aniline blue-eosin (Anilin SIGMA-Aldrich, Eosin Y Merck Hungary) was measured into the Eppendorf tube, 10 μl of semen was measured into the stain, the semen was

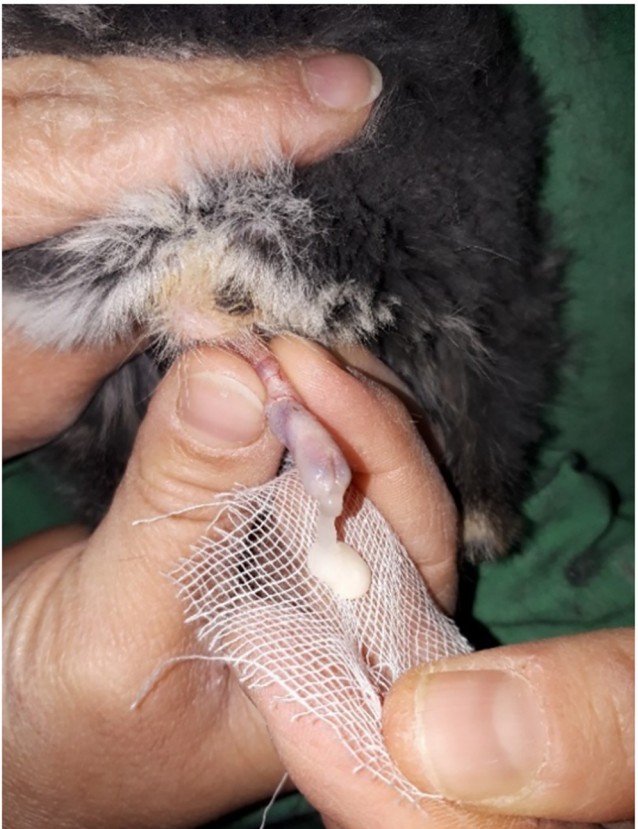

**Fig 1. Collection of sperm sample on a gauze pad whit massage technique base on digital manipulation.**

carefully mixed with the stain. Slides were prepared by smearing 10 µl of the mixture and dried carefully with warm air.

Slides were evaluated microscopically (Zeiss, Axioscope; Carl Zeiss Microscopy GmbH., Göttingen, Germany) using an oil-immersion objective and 1200x magnification.

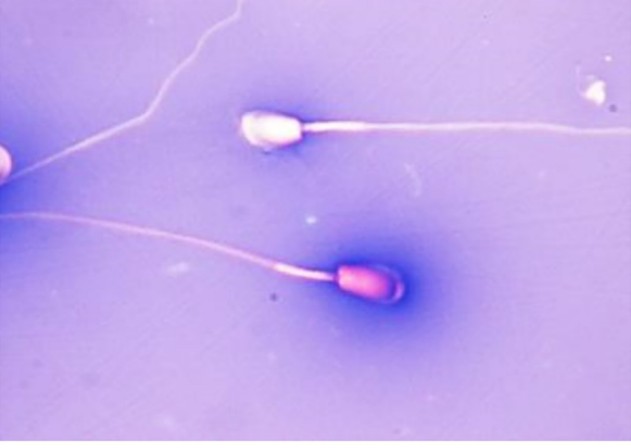

**Fig 2. Differentiating between live (white) and dead (pink) chinchilla spermatozoa in a stained smear using aniline blue and eosin.**

Subsequently, 200 cells per smear were assessed, and the proportion of live, normal morphology to live abnormal and dead cells was determined.

### Ethics statement

The chinchillas were kept in compliance with the animal welfare guidelines defined in the Hungarian Animal Protection Act (Act XXVIII of 1998). Permission for any and all experimental animal research by the research institution was obtained from the National Food Chain Safety Office, Animal Health and Animal Welfare Directorate, Budapest, Hungary (permit number: 13/2015).

### Statistical analysis

Data was analysed using the *Statistica 12.0 program*. For the data expressed and represented as a percentage, statistical analysis was performed following *Arcsine transformation* [22]. As the samples did not show normal distribution, further statistical analysis was performed using the Kruskal-Wallis test, followed by the Kolmogorov-Smirnov two-sample test, while a chi$^2$ probe was used for comparison of sperm collection effectiveness.

## Results

### Semen collection

Actual investigation of the sperm began, once the elaborated sperm collection method had proven to be safe and the animals had become accustomed to it. The success rate of manual semen collection varied between 62.0%-72.6% over the course of the year (Table 1). Regarding the success rate of semen collection attempts between the individual seasons, no variation could be observed. However, approximately 30% of the animals exhibited no response to this collection technique. Those animals that did not respond after 3–4 attempts would have presumably remained unresponsive later as well; consequently, they were excluded from further investigations.

### Sperm concentration

Large individual differences in concentration were found between the semen samples. The average sperm concentration for the whole year was 935 million/ml. The lowest value was measured in spring (102 million/ml), while the highest in autumn (5800 million/ml). Sperm concentrations showed a changing trend over the course of the year (Fig 3).

Regarding seasonal comparisons, concentrations were found to be the lowest in spring, followed by an increasing trend in both winter and summer, with the highest levels in autumn (p≤0.05).

### Sperm motility

For the year in total, rapid, progressive cells comprised the highest proportion in the samples (Table 2), with the highest percentage of them in autumn (65.0%) and winter (70.7%), with,

**Table 1. Seasonal distribution of successful semen collection.**

| Season | Total semen collection attempts | Successful semen collection | Ratio of successful semen collection (%) |
|---|---|---|---|
| Spring (March, April, May) | 168 | 122 | 72.6 |
| Summer (June, July, August) | 187 | 116 | 62.0 |
| Autumn (Sept, Oct, Nov) | 187 | 126 | 67.4 |
| Winter (Dec, Jan, Febr) | 203 | 139 | 68.5 |

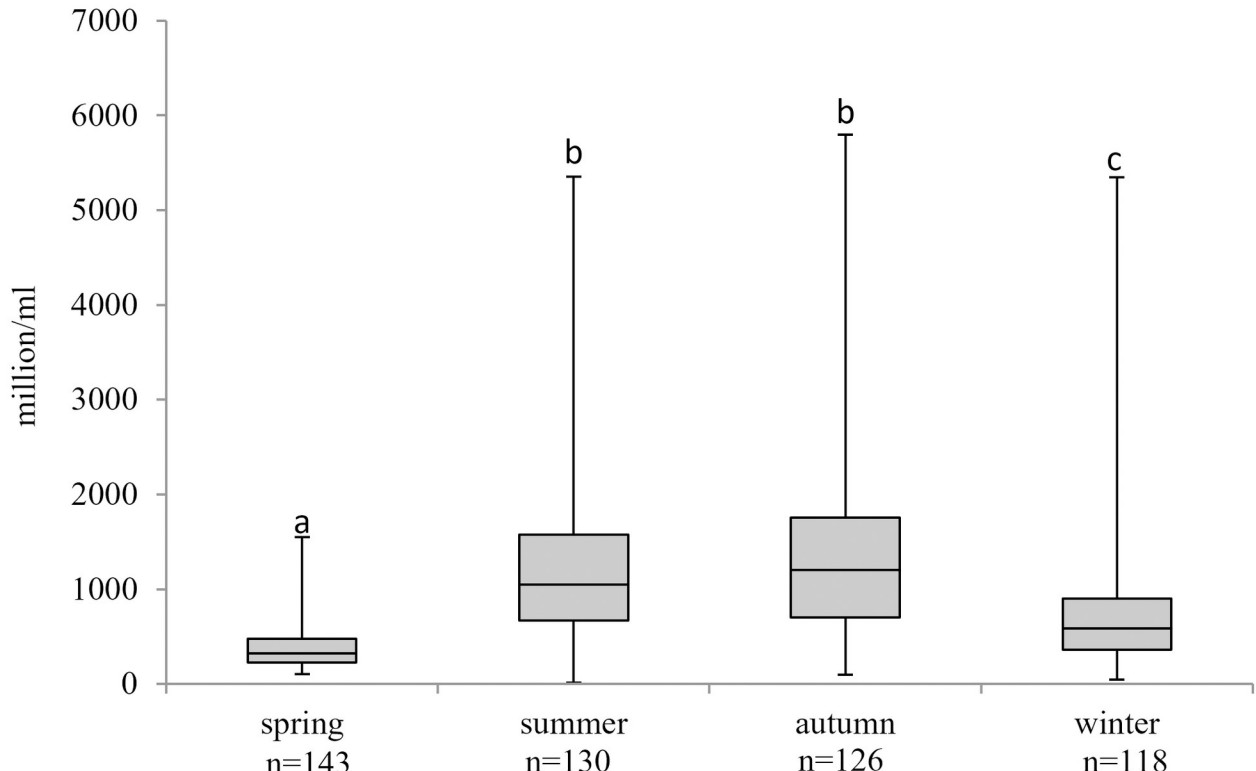

**Fig 3. Seasonal variations in sperm concentration during the study period.** Values in parenthesis represent the number of samples evaluated per season. Different letters indicate statistical differences for sperm concentration. a;b;c: p≤0,05.

notably, the lowest in spring (56.1%). The proportion of medium progressive cells varied between 1.8% and 13.3% over the year, with the highest values in spring (13.3%) and the lowest in autumn (1.8%). The proportion of non-progressive cells varied between 15% and 21.7% over the year, with the higher proportion observed in summer (20.2%) and autumn (21.7%), while slightly lower in spring (15.0%) and winter (15.1%). The proportion of immotile cells varied from 9.4%-15.7%, with their highest average in spring (15.7%) and the lowest in winter (9.4%). Total cell motility was the highest in winter (90.3%), followed by autumn (88.5%) and summer (85.7%), and the lowest in spring (84.3%). For all categories collectively, there was a noticeable difference in cell motility between seasons, with the highest values overall achieved in winter.

**Table 2. Seasonal distribution of motility parameters.**

| Season | Rapid progressive % | Medium progressive % | Non- progressive % | Immotile % | Total motile cells % |
|---|---|---|---|---|---|
| Spring (March, April, May) | 56.1[a] | 13.3[a] | 15.0[a] | 15.7[a] | 84.3[a] |
| Summer (June, July, August) | 59.8[a,b,c] | 5.7[b] | 20.2[b,c] | 14.3[a,b] | 85.7[a,b] |
| Autumn (Sept, Oct, Nov) | 65.0[b] | 1.8[c] | 21.7[b] | 11.5[b] | 88.5[b] |
| Winter (Dec, Jan, Febr) | 70.7[b,d] | 4.8[b] | 15.1[a,c] | 9.4[c] | 90.3[c] |
| p≤0,01 | a-b;c-d | a-b;a-c;b-c | a-b;b-c | a-b;a-c;b-c | a-b;a-c;b-c |

Different letters indicate statistical differences for various motility parameters by season. There is a significant difference between letters connected by hyphens.

## Sperm viability and morphological abnormalities

In smears stained with eosin-aniline blue, the proportion of live, intact cells was above 80%, while the ratios of live, morphologically abnormal and dead cells were 6% and 14%, respectively, on average, for the period we studied (Fig 4). The proportion of live, intact spermatozoa slightly varied throughout the year, similar to cell concentrations.

The ratio of live, intact cells was the lowest in spring (81.09%), no notable difference was found between summer and autumn (84.71% and 83.58%), and the highest was achieved in winter (87.07%). No difference was found in the amount of abnormalities in spring and summer (3.49% and 3.52%), these were followed by a slight decline in autumn (2.82%) and winter (1.76%).

The proportion of each morphological abnormality was represented as a percentage of the total number of cells examined (Fig 5). Examining these morphological abnormalities in more detail, we found that midpiece abnormalities occurred in the highest proportion (0.95%-3.38%), followed by acrosome abnormalities (0.28%-1.38%) in almost all seasons, while head abnormalities showed the lowest ratio (0.01%-0.15%) overall. The percentage of acrosome abnormalities was the highest in the spring (1.38%) and the lowest in winter (0.28%). Conversely, the incidence of head abnormalities was lowest in spring (0.01%), with no differences found in the other seasons. The lowest rate of midpiece anomalies occurred in winter (0.95%) and the highest rate in summer (3.38%). No differences were found in the incidence of tail abnormalities between seasons (0.32%-0.52%).

## Discussion

This study represents the first of its kind to evaluate sperm parameters over the duration of a complete year, collecting the semen of the Chinchilla Lanigera using a massage method. Prior

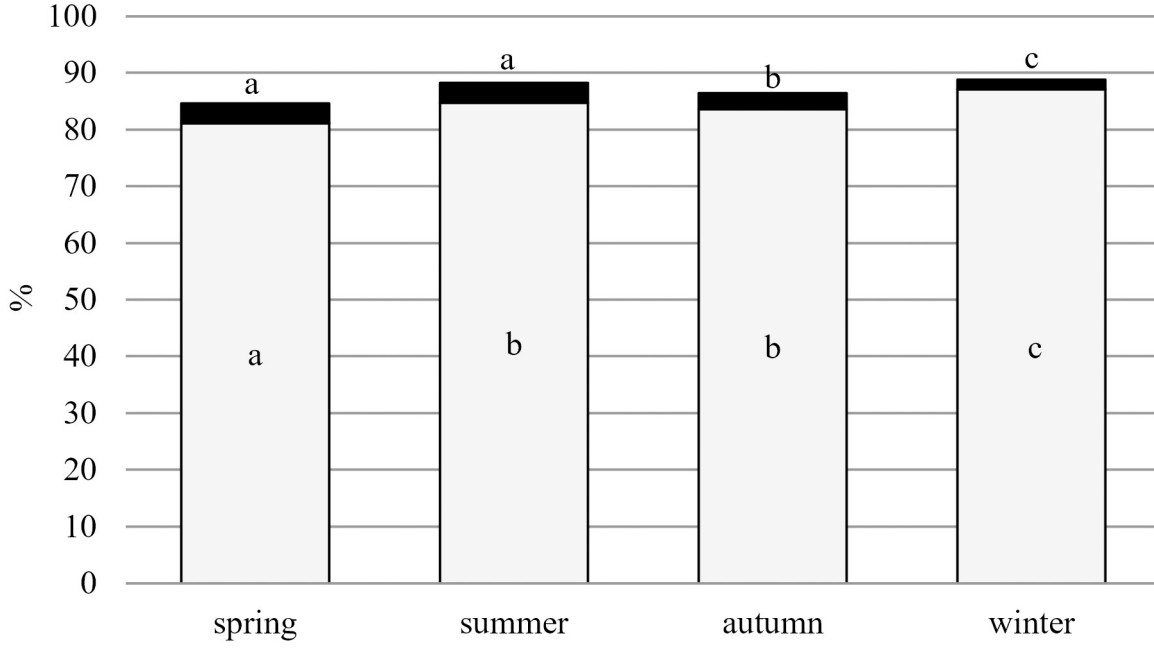

**Fig 4. Seasonal distribution ratio of live, intact and live, abnormal spermatozoa in eosin-aniline blue-stained smears.** Different letters indicate statistical differences for live, intact cells and abnormal cells by season. a;b;c: p≤0,05.

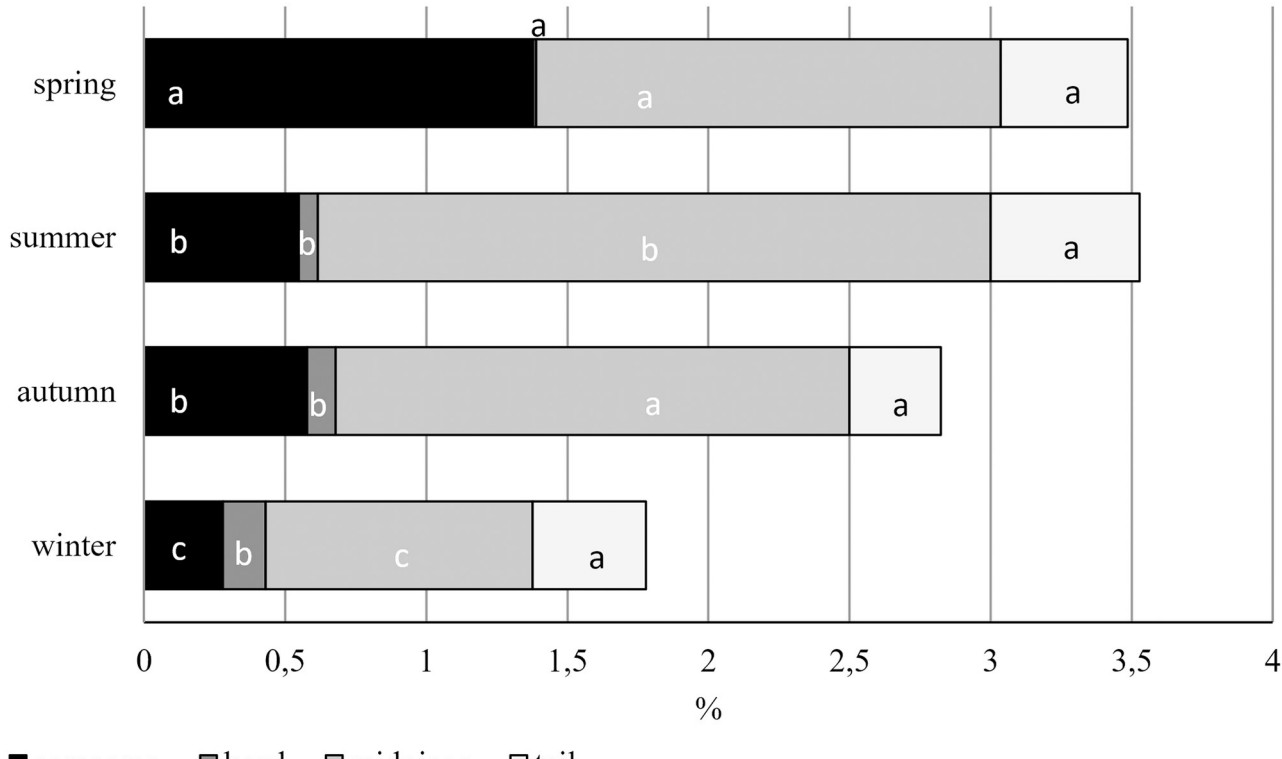

**Fig 5. Detailed seasonal distribution of morphological abnormalities in eosin-aniline stained smears.** Morphological abnormalities are plotted as a proportion of all examined cells. Different letters indicate statistical differences for different abnormalities by season. a;b;c: p≤0,05.

to today, the only documented method of sperm collection used in chinchilla examination had been electro-ejaculation. Digital manipulations were tested on some wild species such as Malaysian estuarine crocodile (*Crocodylus porosus*) with success [23] while in the case of Malayan Pangolin (*Manis javanica*) was less successful [24] According to the results, obtained here, effective collection of semen can be achieved without invasive interventions, using a technique based on digital manipulation, which is a fundamental and necessary step forward in both preserving the species, and in acquiring further data regarding spermatological parameters. One possible disadvantages of the massage technique may be that, while electro-ejaculation is typically able to collect semen from almost all participants involved, only 70% of the cases using the manual technique were successful. However, with the manual collection of semen can be performed several times per week, according to our preliminary tests, but at least twice a week, so overall more semen can be collected per week than with electro-ejaculation. Certainly, through a stricter selection of the males and by further refinement of the method itself, this level of effectiveness can be improved.

Data on the spermatological characteristics of chinchilla is scarce and what does exist ranges widely (e.g., concentration from 1 to 11712 million/ml) [2, 10, 14, 15, 20, 21, 25]. One reason for these erratic results might be due to the reproductive seasonality suggested in earlier examinations, and another may also depend on the different electro-ejaculation techniques applied. In this study, differences were observed between the parameters, not only in the semen samples of individuals, but also between the seasons overall. Some researchers have found definite seasonality in the reproductive performance of chinchillas [6, 26, 27], however,

according to the results of [15], sperm production can be achieved all year round. Based on our research, we also found that semen could be collected from chinchillas at any time during the year; however, at certain times of the year, differences in the individual parameters examined were experienced, thus, some, but not definite, seasonality can also be said to exist.

It must be stated, that the gauze used in Eppendorf tubes during sperm collection retains some of the spermatozoa; therefore, the recorded sperm count shows only the amount available for use in artificial insemination, but not the absolute value. Previous authors have given a wide range of cell concentrations (from 4.19 to 11712 million/ml), and the concentration of the semen we collected, although similarly wide-ranging, falls well into this range (126.1–4448.8 million/ml). According to the results of Busso et al. [10] and Dominchin et al. [15], the concentration of spermatozoa did not change significantly throughout the year. In our study, the highest concentrations were found in winter, while the lowest in spring, with the overall effectiveness of sperm collection being the lowest in August and September. According to Busso et al. [10], the volume of the testes is at its lowest in the middle of summer, while in this study, concentrations of spermatozoa shown to be the highest in summer and autumn, taking into consideration only those animals from which a semen sample could be obtained.

It is known, that fertility correlates most strongly with the sperm motility values of a given specimen, hence, the ability to objectively determine this value may prove to be of great importance. Most of the earlier studies, through subjective methodologies, estimated motility to be above 69%-100% [2, 10, 14, 15, 21, 25], while the results in our analysis, using the CASA system, ranged between only 80%-90%. Thus, this study found a slightly lower level of motility compared to previous, subjective estimates. Moreover, detailed motility parameters (rapid progressive, medium progressive, non-progressive, immotile) were not defined in earlier data.

Few studies are available on sperm viability. In those studies, the viability of spermatozoa varies between 84%-95% [10, 21, 25, 28, 29]. In this present study, the proportion of live, intact spermatozoa was slightly lower (75%-85%); however, this difference might be due to the fact that a separate category for live, but abnormal, cells was created.

No data regarding the types of morphological abnormalities of spermatozoa were found in earlier studies. Niedbala et al. [25] examined sperm in smears stained with eosin-nigrosine, however, only the proportion of live/dead cells was determined. Overall, morphological abnormalities occurred in an acceptably low proportion of the semen samples examined. The highest rate of abnormality was observed in cells with degenerated midpieces (crocked neck, beat, thick, cytoplasmic droplet), which are known as secondary abnormalities, and can typically occur during the handling of the samples.

## Conclusion

Although many relevant questions still remain open, the results of this particular study show that elaborated manual massage is a suitable method for semen collection of Chinchilla Lanigera throughout the year, with spermatological parameters differing slightly among seasons. All things considered, in terms of motility, concentration, viability and abnormal cell ratio, the worst results were observed in spring, while the healthiest were achieved in winter and autumn. Thus, some amount of seasonality may be experienced by using the system demonstrated in our study, however, the qualification parameters measured nonetheless indicate both that semen samples are capable of being collected all year round, and that they are sufficiently suitable for the purpose of fertilisation, including artificial insemination.

## Author Contributions

**Data curation:** Bianka Babarczi.

**Funding acquisition:** Károly Kustos.

**Investigation:** Bianka Babarczi, Árpád Drobnyák, Éva Váradi Kissné, Zsuzsa Szabó, Zsuzsanna Szőke, Barbara Végi.

**Methodology:** Bianka Babarczi.

**Project administration:** Mónika Heincinger.

**Supervision:** Judit Barna, Barbara Végi.

**Writing – original draft:** Bianka Babarczi, Barbara Végi.

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
