## [Decision Letter · Decision Letter 0]

14 Feb 2023

PONE-D-22-30686Elaboration of massage technique for semen collection and examination of semen characteristics in chinchilla (Chinchilla lanigera)PLOS ONE

Dear Dr. Babarczi,

Thank you for submitting your manuscript to PLOS ONE. After careful consideration, we feel that it has merit but does not fully meet PLOS ONE’s publication criteria as it currently stands. Therefore, we invite you to submit a revised version of the manuscript that addresses the points raised during the review process.

ACADEMIC EDITOR:All the three external reviewers have provided valuable comments. The detailed  massage technique should be described, including the protocol for examination of semen characteristics in chinchilla (Chinchilla lanigera). All the questions and concerns from the three viewers should be treated seriously. A list of changes is needed for review. 

We look forward to receiving your revised manuscript.

Kind regards,

Wan-Xi Yang, Ph.D.

Academic Editor

PLOS ONE

Journal Requirements:

This study was supported by the Operational Programme of Economic Development and Innovation / Hungary (Project code: GINOP-2.1.7-15-2016-02232).

Reviewers' comments:

Reviewer's Responses to Questions

**Comments to the Author**

1. Is the manuscript technically sound, and do the data support the conclusions?

Reviewer #1: Yes

Reviewer #2: Yes

Reviewer #3: Partly

2. Has the statistical analysis been performed appropriately and rigorously? 

Reviewer #1: Yes

Reviewer #2: Yes

Reviewer #3: Yes

3. Have the authors made all data underlying the findings in their manuscript fully available?

Reviewer #1: Yes

Reviewer #2: Yes

Reviewer #3: Yes

4. Is the manuscript presented in an intelligible fashion and written in standard English?

Reviewer #1: Yes

Reviewer #2: Yes

Reviewer #3: Yes

5. Review Comments to the Author

Reviewer #1: Overall, this is a clear, concise and structured manuscript. Apart from the theriogenology studies, further investigation on the reproductive rate of chinchillas either in natural or captive could be interesting to be explored.

Reviewer #2: Introduction :

- lacks introduction to how the massage technique was conceived, please add a few line and cite relevant references

M&M:

This section should have more detailed on "technique for semen collection and examination of semen

characteristics". Currently the manuscript lack depth for understanding of the technique.

Please include details and figures on:

1. There was no elaboration on how the chinchilla was restrained

2. The area of anus massage and the entire penile length on a chinchilla to aid reader to understand the procedure

3. Photomicrograph of the sperm

Results:

Table 1 should be improved - column 4 stated as "%". What does this imply?

Table 2: Row 5 - I am not sure how this data is presented?

Reviewer #3: The chinchilla is an endangered species in the wild, so some research on their reproduction is very important for the species perpetuation. This manuscript developed a manual technique for semen collection which met all animal welfare requirements. In addition, the basic parameters of sperm obtained by this method was determined. This study has definite value for reproduction and animal welfare of the chinchilla. However, I have some major concerns and minor concerns for this manuscript.

Major concerns:

1. In this study, the authors just analyzed the parameters of sperm obtained by the manual technique and some spermatological parameters exhibit a larger fluctuation in the different times of the year. At the same time, this manual technique works for only 66% of chinchillas. So I suggest if the authors want to show the manual technique is a better way for semen collection than the common electro-ejaculation methods, they can analyze the parameters of sperm collected by electro-ejaculation methods. And through comparison and analysis, not only the advantages of the manual technique, but also the development of this technique will show up better.

2. The authors analyzed many spermatological parameters in this study. However, what’s the relationship between the results of these parameters and the manual technique developed in the study?

3. About the manual technique, the authors did not write it clearly. I suggest it can be descripted in more detail in the methods part.

4. The discussion part is too simple. I suggest the authors performed a profound discussion on the importance of this manual technique and meaning for the reproduction of chinchillas.

5. Several brackets are italic incorrectly, such as (Chinchilla lanigera) , (Table 1) , (Fig.1)……

6. The references of this manuscript are so old and I suggest to add some new references.

6. PLOS authors have the option to publish the peer review history of their article (what does this mean?). If published, this will include your full peer review and any attached files.

Reviewer #1: **Yes: **Dr Mohd Iswadi Ismail

Reviewer #2: **Yes: **Wan Nor Fitri Bin Wan Jaafar

Reviewer #3: No

---

## [Author Response · Author response to Decision Letter 0]

2 May 2023

Reviewer #1: 

Overall, this is a clear, concise and structured manuscript. Apart from the theriogenology studies, further investigation on the reproductive rate of chinchillas either in natural or captive could be interesting to be explored.

Thank you for your opinion. The topic you raise is indeed very interesting. The ultimate aim of our studies is to develop artificial insemination of chinchillas, the first step of which is to develop an effective, animal friendly sperm collection.

Reviewer #2: Introduction :

- lacks introduction to how the massage technique was conceived, please add a few line and cite relevant references

Thank you for your comment. The digital manipulation or massage technique for sperm collection usually is commonly used in dogs, pigs and poultry among other species. In this regard, we have added references to the introduction section. Massage method for sperm collection has not yet been described in Chinchilla, we are the first to have developed it.

M&M:

This section should have more detailed on "technique for semen collection and examination of semen characteristics". Currently the manuscript lack depth for understanding of the technique.

Please include details and figures on:

1. There was no elaboration on how the chinchilla was restrained

2. The area of anus massage and the entire penile length on a chinchilla to aid reader to understand the procedure

3. Photomicrograph of the sperm

We have added a more detailed description of the massage technique in the Material and Methods, including a picture about sperm collection and spermatozoa smears. 

Results:

Table 1 should be improved - column 4 stated as "%". What does this imply?

In the fourth column, we have added the name of the column to make it easier to understand.

Table 2: Row 5 - I am not sure how this data is presented?

The data in row 5 present significant differences. We have added the following sentence to the title of the table, which may help you to understand it better: “There is a significant difference between letters connected by hyphens.”

Reviewer #3: 

The chinchilla is an endangered species in the wild, so some research on their reproduction is very important for the species perpetuation. This manuscript developed a manual technique for semen collection which met all animal welfare requirements. In addition, the basic parameters of sperm obtained by this method was determined. This study has definite value for reproduction and animal welfare of the chinchilla. However, I have some major concerns and minor concerns for this manuscript.

Major concerns:

1. In this study, the authors just analyzed the parameters of sperm obtained by the manual technique and some spermatological parameters exhibit a larger fluctuation in the different times of the year. At the same time, this manual technique works for only 66% of chinchillas. So I suggest if the authors want to show the manual technique is a better way for semen collection than the common electro-ejaculation methods, they can analyze the parameters of sperm collected by electro-ejaculation methods. And through comparison and analysis, not only the advantages of the manual technique, but also the development of this technique will show up better.

Thank you for your comments. We put a detailed description of the developed new technique in the M&M section of the manuscript. The primary aim of our work was not to compare electroejaculation with the massage technique, but to develop a technique in a new approach and checking whether it was useable to get semen with good quality all year round. We cited electroejaculation method in the introduction section because it has been the only technique for collection semen in chinchillas so far. Since there were no data on the quality of semen from the massage technique until now, we compared them with data from the available literature. We are not trying to prove that the massage technique is much better than electroejaculation, but rather to see if this method can work without anaesthesia, expensive facilities and torturing animals. Anyway, the new technique has another advantage, namely to obtain higher amount of spermatozoa per week in total, even though only 60-70% of males respond to stimulation, since it is useable several times a week compared to the electroejaculation which can be used only once a week or every two weeks.

2. The authors analyzed many spermatological parameters in this study. However, what’s the relationship between the results of these parameters and the manual technique developed in the study?

Since earlier literature data on spermatological parameters show large deviations, we could only conclude that the values obtained with the manual technique are similar and fit to the available data. Therefore, the technique can be used safely for sperm collection and insemination throughout the year, which is the basis for the development of artificial insemination of the species.

3. About the manual technique, the authors did not write it clearly. I suggest it can be descripted in more detail in the methods part.

We have added a more detailed description of the massage technique in the Material and Methods, including a picture about sperm collection and spermatozoa smears. 

4. The discussion part is too simple. I suggest the authors performed a profound discussion on the importance of this manual technique and meaning for the reproduction of chinchillas.

We discuss in details the results and the analysis of our study over two pages with several available references, thus do not think that the section is unsatisfactory for the professionals. We added some new statements to the section regarding the further advantage of the method.

It includes all the new results, such as the fact, that only electroejaculation has been used to collect sperm until now, and only subjective estimation of motility has been used so far. Thus, we are the first to use CASA to test motility, to show first the types of sperm abnormalities in the species. Furthermore we were the first to describe a novel, effective massage technique in chinchillas. 

5. Several brackets are italic incorrectly, such as (Chinchilla lanigera) , (Table 1) , (Fig.1)……

Thank you for your comment, it has been corrected in the text.

6. The references of this manuscript are so old and I suggest to add some new references.

The reviewer is right that the literature used is not from today and not very much of it. However, this is because the last article on chinchilla reproduction was published in 2014. Unfortunately the researches of reproductive biology of chinchillas seems not to be a very popular topic, recently chinchillas are more commonly studied by researchers in human medical diagnostics from different aspects. If you know any recent publications, we would welcome your recommendation.

---

## [Decision Letter · Decision Letter 1]

6 Jun 2023

PONE-D-22-30686R1Elaboration of massage technique for semen collection and examination of semen characteristics in chinchilla (Chinchilla lanigera)PLOS ONE

Dear Dr. Babarczi,

Thank you for submitting your manuscript to PLOS ONE. After careful consideration, we feel that it has merit but does not fully meet PLOS ONE’s publication criteria as it currently stands. Therefore, we invite you to submit a revised version of the manuscript that addresses the points raised during the review process.

Please pay attentions to the reviewer's comments. The figures should be labled accordingly. 

We look forward to receiving your revised manuscript.

Kind regards,

Wan-Xi Yang, Ph.D.

Academic Editor

PLOS ONE

Journal Requirements:

Reviewers' comments:

Reviewer's Responses to Questions

**Comments to the Author**

1. If the authors have adequately addressed your comments raised in a previous round of review and you feel that this manuscript is now acceptable for publication, you may indicate that here to bypass the “Comments to the Author” section, enter your conflict of interest statement in the “Confidential to Editor” section, and submit your "Accept" recommendation.

Reviewer #1: All comments have been addressed

Reviewer #3: All comments have been addressed

2. Is the manuscript technically sound, and do the data support the conclusions?

Reviewer #1: Yes

Reviewer #3: Yes

3. Has the statistical analysis been performed appropriately and rigorously? 

Reviewer #1: Yes

Reviewer #3: Yes

4. Have the authors made all data underlying the findings in their manuscript fully available?

Reviewer #1: Yes

Reviewer #3: (No Response)

5. Is the manuscript presented in an intelligible fashion and written in standard English?

Reviewer #1: Yes

Reviewer #3: (No Response)

6. Review Comments to the Author

Reviewer #1: All comments provided by the author(s) are acceptable. However, there are a few improvement need to be made;

1. Inconsistent reference list

- There is inconsistent in writing the reference list. Author(s) should check the journal's format

2. The Fig 1 & Fig 2 should be labelled

-A point of interest in the Fig 1 & Fig 2 should be precisely labelled

Reviewer #3: (No Response)

7. PLOS authors have the option to publish the peer review history of their article (what does this mean?). If published, this will include your full peer review and any attached files.

Reviewer #1: **Yes: **Mohd Iswadi Ismail

Reviewer #3: No

---

## [Author Response · Author response to Decision Letter 1]

13 Jul 2023

Dear Reviewer,

thank you for your work, the change is you have requested, have been made.

1. Inconsistent reference list

- There is inconsistent in writing the reference list. Author(s) should check the journal's format

We check the format references, and we have corrected the incorrect information and the inconsistencies.

2. The Fig 1 & Fig 2 should be labelled

-A point of interest in the Fig 1 & Fig 2 should be precisely labelled

Based on your request we proposed the following changes to the title of the figures: 

Fig 1. Collection of sperm sample on a gauze pad whit massage technique base on digital manipulation

Fig 2. Differentiating between live (white) and dead (pink) chinchilla spermatozoa in a stained smear using aniline blue and eosin

---

## [Editor Report · Decision Letter 2]

9 Aug 2023

Elaboration of massage technique for semen collection and examination of semen characteristics in chinchilla (Chinchilla lanigera)

PONE-D-22-30686R2

Dear Dr. Babarczi,

We’re pleased to inform you that your manuscript has been judged scientifically suitable for publication and will be formally accepted for publication once it meets all outstanding technical requirements.

Kind regards,

Wan-Xi Yang, Ph.D.

Academic Editor

PLOS ONE
---

## [Editor Report · Acceptance letter]

22 Aug 2023

PONE-D-22-30686R2 

Elaboration of massage technique for semen collection and examination of semen characteristics in chinchilla (*Chinchilla lanigera*) 

Dear Dr. Babarczi:

I'm pleased to inform you that your manuscript has been deemed suitable for publication in PLOS ONE. Congratulations! Your manuscript is now with our production department. 

Kind regards, 

on behalf of

Dr. Wan-Xi Yang 

Academic Editor

PLOS ONE